# The Potential of Using an Eye Tracker in Architectural Education: Three Perspectives for Ordinary Users, Students and Lecturers

**Marta Alina Rusnak \*** and **Mateusz Rabiega**

Faculty of Architecture, Wrocław University of Technology, 50-370 Wrocław, Poland; mateusz.rabiega@pwr.edu.pl
*   Correspondence: marta.rusnak@pwr.edu.pl

**Abstract:** The aim of this paper is to discuss the potential of eye trackers as tools providing diversified support for the architectural education of future designers. The possibility to track eye movement guaranteed by this group of devices enables an extension of knowledge on the non-professional perception of architectural creations. It also allows people to monitor progress while learning and verify a project's assumptions as well as provide lecturers with an opportunity to optimize didactic methods. The paper includes authors' ideas for modifications of teaching methods applied at technical universities. It is a result of an analysis of research related to the perception of urban designs and architectural objects—research during which noticeable differences were observed between how experts and non-professionals perceive these structures. What also contributed to the contents of this paper was a comprehension of the wide range of eye tracking studies examining the level of acquisition of specialist professional skills. The presented ideas are also based on the analysis and adaptation of eye-tracking research conducted by scientists within other areas of life.

**Keywords:** eye tracking; architectural education; participation; self evaluation; building design; human reactions; experiments

## 1. Introduction

The deliberations presented in this paper stem from the surprising results of another research study in which the opinions of professionals and visual reactions of non-professionals were compared in the context of choosing the color of a logo placed in front of a UNESCO-listed museum building [1]. In order to interpret the obtained results, this research study was followed by an in-depth literature study. This paper includes references to opinions of experts who noticed a need for thorough observation of the visual behavior of users of architecture in order to boost the pro-social aspect of architectural education. The authors were interested in examples of how eye trackers are used in the widely understood field of architecture and how experts and laypeople perceive various visual stimuli as well. They were also keen on finding ways in which education may be improved by means of eye trackers. An important part of this paper is the presentation of the first attempts at implementing an eye tracker as one of didactic tools at the Faculty of Architecture of Wrocław University of Science and Technology. The aim of the paper is to analyze the possibilities of using an eye tracker in improving the course of architectural education as well as presenting overall social benefits from their common application at universities educating future designers.

By combining their knowledge and experiences, the authors aimed to put their thoughts in order and determine the range and potential direction of further research. The paper is meant to make the readers aware of the issue at hand, review the available eye-tracking techniques and their applications to date, and describe the first attempts at implementing eye trackers in education. It is vital to encourage academics to search for

new ways of improving the process of architectural education, including those that employ biometric devices.

### 1.1. The Need to Verify Views

Many architects, conservators, museologists, theorists of architecture and specialists in different fields have been looking for a way to register how a particular object or space is perceived by ordinary onlookers. They want to find answers to questions related to both practical skills (for instance, whether or not a museum visitor actually reads the descriptions under the exhibits) and theoretical knowledge (for example, how different architectural compositions are perceived).

One example of a search for a way to examine the visual behavior of visitors is a statement of Paweł Kowal, the co-creator of the Warsaw Rising Museum, who declared that he would like to install video cameras on the heads of visitors in order to observe where they stop and which parts of the exhibition they avoid. What he wanted was to diagnose both the drawbacks of the exhibition as well as those places that draw particular attention [2].

Another group of professionals that is highly interested in the visual responses of people are conservation officers and architects who deal with interpretations of the past [3,4]. An example of such interest may be seen in considerations of anastylosis, which is a reassembly of damaged relics of the past [5]. When it is decided that a given object should be reassembled, one needs to choose the color and texture of the new additions so that they can be differentiated from the original parts of the relic. In the end, although there are documents and guidelines [6], designers and conservators have to make their own decisions about the level of contrast between the old and the new elements.

Apart from such very practical issues that are important to many contemporary designers, many historians of architecture touched on the topic of how people perceive objects raised many years ago. Nikolaus Pevsner and Juliusz Żórawski tried to explain to their readers what the dominant ways of looking at the structure of a gothic church are [7] and how one perceives the depth of a medieval cathedral [8]. The history of architecture offers plenty of such perceptual riddles. A particularly large number of them is related to the epoch of baroque when the creativity of details used by architects was meant to achieve a specific effect—what contributed to it was the use of massive objects and convex shapes, the optical games that juxtaposed different directions inside a building, and the contrast between the bright and dark areas [9]. As many historians of art noted, David Watkin and Nikolaus Pevsner among them, it was important to steer the visitor to discover parts of an observed object in a particular order [10].

The abovementioned examples show that the issue of visual perception of various architectural elements is continuously drawing the attention of scientists and researchers. The need to develop curiosity in architects about how different people perceive their surroundings seems both essential and natural. The Polish system of architectural education only indirectly addresses this need. In the Ordinance of the Minister of Science and Higher Education from the 18 July 2019, one may find the following statement: "The graduate knows and understands relations between man and architecture and between architecture and the surrounding environment, and the necessity to adapt architecture to human needs and scale" [11]. Among various needs, one should not neglect the diversity relating to visual needs. Architects ought to verify their assumptions by checking whether their directed and aesthetically formative education allows them to correctly diagnose and satisfy the needs of their recipients.

The ideas mentioned above seem so basic that they should not raise any doubts. However, it is possible to notice the reluctance of a large number of professionals toward the participatory and pro-social nature of some architecture-related research [12].The results mentioned at the beginning [1] and the authors' other data collected during eye-tracking tests regarding anastylosis, looking at museum interiors, and the perception of depth in churches [13–15] suggest that such research should not be treated as a fad or a scientific

whim [16], but rather a necessary act of cleansing the didactic contents included in the syllabuses of various architecture faculties from theories constructed on the basis of sheer intuition. Other scholars have also noticed the discrepancies in this respect [17]. One of the mentioned researchers even claims that such a state of things results in a pathological "over-specialization" that ignores the needs "of ordinary people", and repeats rules that have not been verified for years [12]. As a consequence, various spaces, interiors and details are designed in accordance with rules that are deemed fundamental but that are radically different from the actual needs and feelings of their everyday users [12]. From a sociological perspective, it should come as no surprise that experts, being better educated, find it easy to consider the opinions of others as less meaningful. This mechanism is confirmed by research and social practice [18,19]. Very often, the reason for ignoring the opinions of others is their low level of education and lack of competence when it comes to describing a given problem [20]. The results of talks and the answers given by ordinary citizens in surveys are often so diverse that even those experts who would like to take their public opinion into account find them hard to interpret.

If we define the society as the employer and the architects as employees responsible for executing specific tasks, it seems that the latter should aim at rationally satisfying the needs of the community by observing behaviors of individuals that the community consists of. "This is of crucial importance regarding the respective needs to be satisfied, even if they have not yet been revealed or discovered" [21].

The authors suggest that in time, eye trackers—as devices allowing the measurement of both deliberate and instinctive visual responses—could become a solution to these problems. The registration of what non-professionals look at in relation to different designing issues could potentially become a sort of "corrective lenses" to more than one "architectural myopia" [12].

*1.2. Eye Tracking*

In order to make it possible for the readers to fully understand the presented ideas, the authors see it as their duty to explain what eye tracking is. Eye tracking consists of following eye movements. Modern technologies allow for nearly completely non-invasive registering of how different people look at the presented objects. Recordings can be performed with the use of three groups of devices [22,23]: first, stationary eye trackers (Figure 1) intended for the analysis of such 2D stimuli as photos or videos. Among these devices, one may find those that are permanently connected to a screen and those that are compatible with other methods of presenting stimuli. The second group consists of mobile eye trackers worn in the same way as glasses or as a forehead band. These devices make it possible to carry out tests both indoors as well as outdoors and the way in which a participant perceives their surroundings is, in this case, recorded in the video registered by a small camera. Another group of eye trackers, which has recently been introduced on the market, are devices integrated with VR goggles, e.g., Tobi XR, HPReverb G2. Thanks to them, it is possible to base the studies on spherical stimuli: photos, videos or even games in the form of special spaces, e.g., virtual laboratories [24]. All devices, irrespective of whether the study participant was watching a video, a photo or a real item, record the places and moments when their sight was fixed on them. Appropriate software divides this process into fixations—short moments of maintaining the visual gaze [22] and saccades—fast movements/switches between the locations on which the observer's visual attention is concentrated [22]. Such division of the eye tracking process enables precise mathematical interpretation, on the condition that the methodology is properly planned and the data are carefully gathered [25].

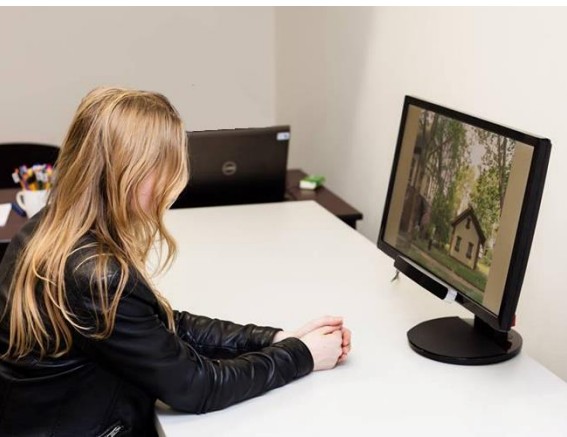

**Figure 1.** A student testing the functionality of a stationary eye tracker (photo from studies in 2017).

## 2. Materials and Methods

### 2.1. Methods

Diverse experiences related to eye tracking made the authors decide that it is necessary to analyze different ways in which such devices could be used, not only in research, but also as tools facilitating the process of architectural education. The analysis was divided into stages, the first of which consisted in putting in order the experiences of the authors, all the previous research and the potential range of future experiments. It was also important to search for other eye-tracking tests in different fields, which could examine the process of skill acquisition.

The second part of the paper includes a synthesis of the authors' ideas as to when, who and how might apply eye trackers in order to improve the process of broadly understood architectural education—and, in consequence, improve the quality of design services rendered by architects. In this part, the authors try to briefly point out the architectural skills of students and the didactic skills of university lecturers that could benefit from the implementation of eye trackers.

### 2.2. Materials—Scientific Search

#### 2.2.1. Eye Tracking and Architecture

Research done so far on eye trackers and the range of ways in which such tools are currently employed show that using an eye tracker for teaching purposes and for discovering the way of perceiving art and building environments is not completely new. Several researchers have studied how viewers look at paintings [26], while others have made attempts to discuss the potential of implementing an eye tracker into the process of managing museum exhibitions [14,27–29] as well as verifying theories referring to the history of architecture [15] and the subject of perception of historical and urban heritage [30–33]. Experts have also analyzed ways of looking at garden and landscape architecture [34–38]. The eye tracker was also used for improving the method of designing fire-safety marking systems [39], optimizing the ergonomics of interiors or examining consumer behavior in buildings devoted to commerce [40]. Tests have also been conducted to determine the features that are most favorable when designing architectural and urban areas that would prove visually attractive to their users [41,42]. An interesting aspect of eye-tracking research that might influence the way urban areas are designed are the tests related to visual behaviors of drivers and cyclists [43].

#### 2.2.2. Eye Tracking and Education

In the past, eye trackers were used in multiple ways for broadening knowledge on multiple aspects of the teaching process and discovering the way for achieving fluency in performing various professional obligations [44]. The most interesting research was conducted with regard to diagnosis and improvement of professional skills of school

pedagogues [45], doctors and diagnosticians of various medical specializations [46–48], criminologists [49], airplane pilots [50], drivers [51], and navigators of sea vessels [52].

Research may also be used to diagnose typical skills possessed by beginner and advanced programmers [53].

One of the most interesting eye-tracking tests done so far touched on the ability of professionals and students at various stages of education to interpret aerial photographs of archaeological sites [54].

### 2.2.3. Architectural Skills That Could Be Assessed by Means of an Eye Tracker

All architects need to know how to read and interpret various visual data. It is particularly important when they are to assess someone else's work. For example, they might become responsible for checking building permit designs for mistakes (for their co-workers or as civil servants) or for analyzing graphic data, such as plans, projections or cross sections prepared by students during their university education. Just like other groups of professionals, at various stages of their education, architects solve job-specific "perceptual tasks" [55]. The degree to which they have mastered these skills is difficult to determine, especially when the aim of such evaluation is not only to assess, but also to identify and remove the problems that led to mistakes. The lack of a method allowing a quick diagnosis of such issues makes it impossible to address individual needs, but also excludes appropriate diagnosis of the elements that require more attention in regard to particular groups and stages of education.

Among many such skills, one should distinguish the absolutely basic ones, related to being able to understand and make architectural drawings. Other such issues include being able to anticipate how people would find their way inside designed structures and interiors. For example, will it prove easy for them to find the entrance and understand how to access the garage? Some of these problems may affect not only the comfort of the future users, but also their careers or even safety. If a building's layout does not help users to navigate it, they might have trouble finding a restroom or getting to the check-in desk at an airport on time; it might lengthen the time needed to evacuate the building or for medical personnel to arrive and provide help to a person needing it.

Visual issues become even more complex when an architect deals not with a new building, but an already existing one. They have to be capable of pinpointing the most important features of the structure and predicting how added or modified elements might influence the perception of the whole. If interventions are made in a historical building, the visual interactions caused by the combination of the old and the new parts become extremely important. In such a situation, an architect should know what actions will sustain the heritage of the given place, what will encourage visitors to understand the object's value, and what measures would only do harm.

An architect should also recognize the fact that recipients of architecture are incredibly diverse and that for various reasons—for example, because of some disability—perceive a given space from different perspectives. Very specific tasks also await those experts who become the teachers of next generations of architects. For them, it is essential to apply suitable didactic tools if they want to achieve high standards of education. In the case of studies related to architecture, the majority of presented information is accompanied by visual stimuli, such as photos, schemata or sketches, and the use of eye trackers could ensure that these didactic materials serve their purpose as best as possible.

### 3. Results

*3.1. Eye Trackers and Educating Architects—Authors' Experiences So Far*

Works on the implementation of eye trackers into architectural studies at the Faculty of Architecture of Wrocław University of Science and Technology began in 2014. Although quite a few years have passed, this technology still has not been fully introduced into scientific research and education. So far, the contact with eye trackers has been limited to workshops and training sessions available to a small number of participants. Students

of both master's and doctoral programs were invited to take part in tests. Such a form of gaining new experience and discovering new technologies was very popular among them.

The students had a chance to get acquainted with all three types of eye trackers: the stationary ones, portable ones, and the ones connected to the VR environment. Eye trackers made by SMI and Tobi were available during the research. Due to the time-consuming nature of the analytical process related to interpreting the data obtained by means of portable eye trackers and those linked with VR, the students did not have a chance to apply those two types in their own tests.

Due to the lack of a permanent laboratory, a limited number of rented devices, and the necessity to give everyone individual training, only 20 students took part in the last two experimental sessions in 2017 and 2018—mainly students of the ArcHist Architecture History Research Group.

In total, 51 future architects between the ages of 19 and 27 (22–23 years old on average) had direct contact with the devices and their software. The participants of these activities consisted mostly of senior undergraduate students of architecture and those students in the master's program who had decided to specialize in architectural design in a historical context. Some of them, in cooperation with the author, prepared their own experiments, one of which is shown in Figure 2. Two reports, automatically generated by BeGaze software, relate to the trial registrations that were conducted when the students were being trained on how to operate the stationary SMI eye tracker. Due to the inhomogeneous nature of the tested group and varying circumstances of the experiments, the results of these registrations could not be compared and analyzed. Experiments of this kind, forming part of university classes, would complement the education in the field of aesthetics and museum studies. Only a few of the students were involved throughout the entire research process, as a result, becoming co-authors of future papers.

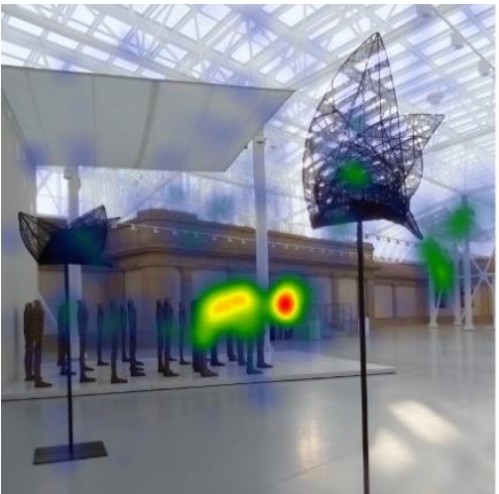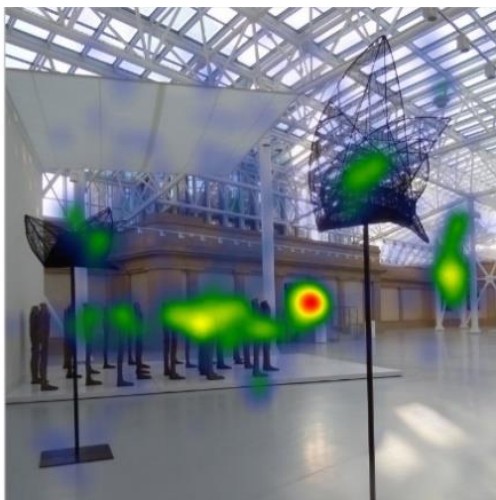

**Figure 2.** Heatmaps obtained after performing simple tests with the participation of MA and PhD students referring to the issues connected with visual perception of museum exhibitions in relationship to the changing architectural environment and artefacts—courtyard of the Four Domes Pavilion in Wrocław (BeGaze Heatmaps on the stimuli prepared by M. Rusnak and ArcHist 2017).

The research that the author conducted and supervised has so far touched on typical design-related issues, such as adaptation and expansion of historical monuments (Figure 3), analysis of perception of museum spaces, conservatory issues, and ways of setting standards for marketing activities in historic surroundings; however, others have also dealt with theoretical issues related to the perception of religious interiors [56].

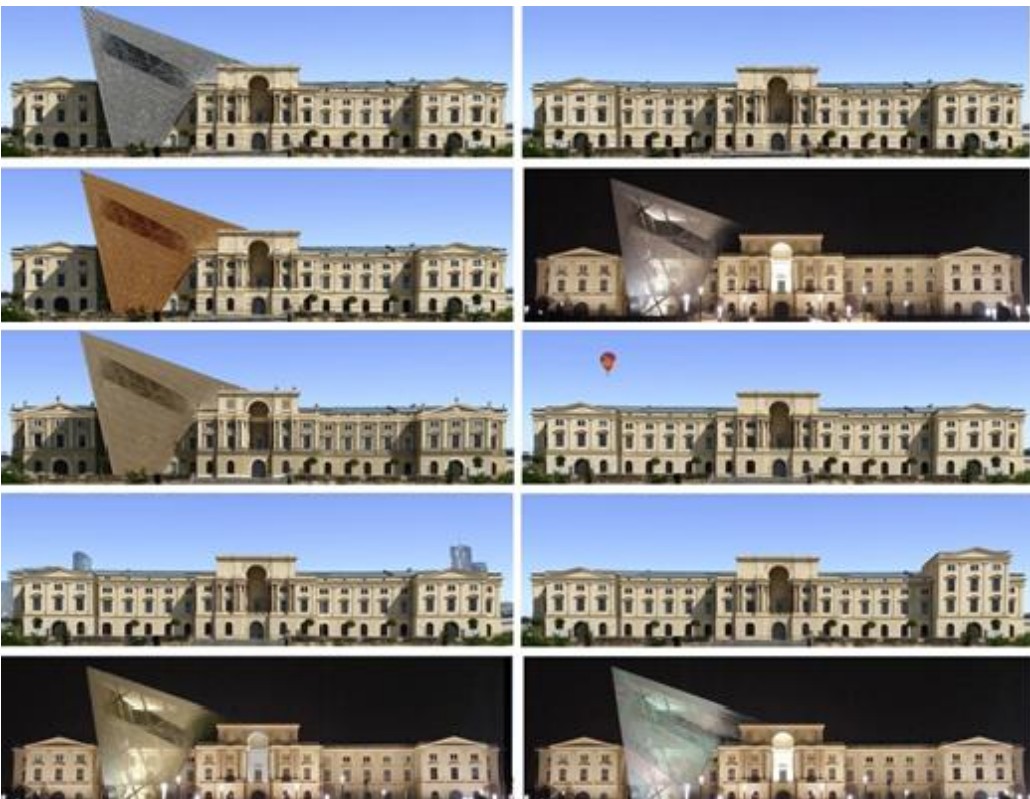

**Figure 3.** Variant studies on the perception of different architectural concepts, including conscious design of illumination, in the scale of a single building seen from the outside [57,58]. In the end, only the first four visualizations were used in eye-tracking research conducted in 2017. The full compilation of these images indicate a wide range of problems and ideas that caught the attention of the students.

In the authors' opinion, the experiments stimulated the students' awareness and critical thinking. Application of an eye tracker was seen by them as a chance to develop themselves and to go on a unique adventure that they designed themselves. They found it intuitive to master this new technology, and that refers to all three types of eye trackers, to the research procedures and to the data interpretation.

The most creative part was not the stage of collecting and interpreting the data, but the time spent on discussions and preparation of the tests. At that time, the students had a chance to make use of their knowledge and various individual experiences in order to suggest phenomena that might prove worthwhile to be researched. During such meetings, the participants tried to find specific objects, quotes or theories that could become a starting point for an experiment. Because of that, the students' propositions were related to a wide range of issues, from adapted interiors to interventions affecting entire buildings or even parts of a city. The discussions touched on how one's perspective of architecture may be altered by such factors as a change of color of some details, time of day and year, different scale or amount of architectural detail, or luminance; they were also related to ways in which one perceives symmetrical and asymmetrical compositions or to the effectiveness of different ways of hiding new objects in the historical tissue of a city (Figure 4). Due to the limitations mentioned before, only some of these issues could be researched. For some of them, the students and their supervisors prepared visualizations and photomontages (Figure 3). Each image was accompanied by a hypothesis; for example, it was expected that a less contrastive element will draw less attention, that an object taking the form of a cuboid covered in vegetation will most likely be unnoticed by onlookers, or that a particular type of illumination will emphasize the importance of a contemporarily added structure. Not all such hypotheses were positively verified. Unfortunately, due to financial shortages,

lack of time and the used methodology, only some of the images could be used in research involving non-professionals.

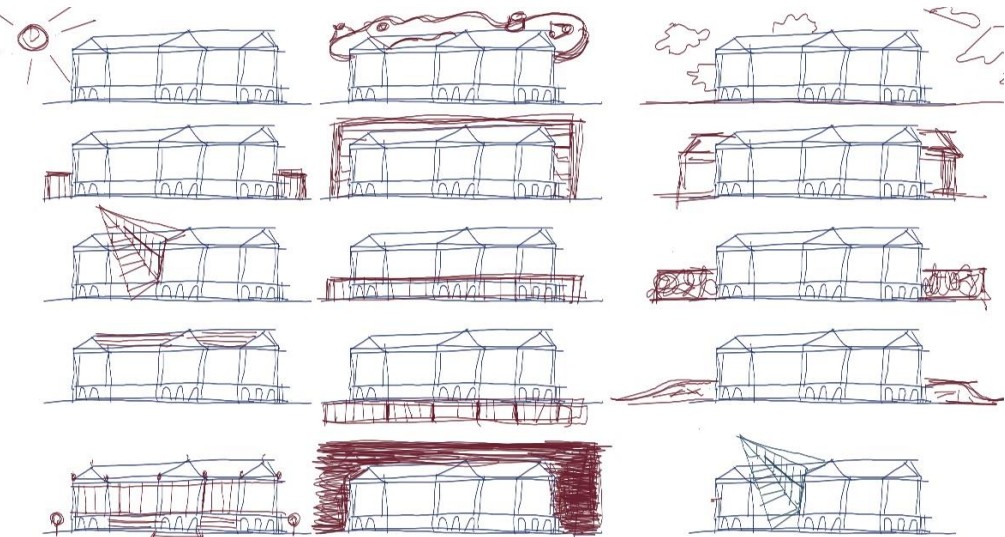

**Figure 4.** Sketches resulting from a meeting with students. They show various ideas of issues that the students wanted to test by means of an eye tracker. The wide range of presented ideas reflects the astonishingly dynamic discussions that occurred during the meetings.

It is worth noting that students excelled at interactions with non-professionals taking part in the tests. It was very stimulating for them to see how many different people participated in the experiments and how varied their perceptions of the visual stimuli were.

If the circumstances allowed, university lecturers who were interested in getting to know this new research device were also invited to the tests (Figure 5). Fifteen academics took part in this form of getting acquainted with new didactic possibilities. In contrast to the students, most of them not only saw the research potential of eye trackers, but also expressed doubts about the application of such devices in regular architectural and conservatory education. The participants frequently asked about the purpose of the tests and expressed concern whether testing non-professionals might lead to such people making decisions on the appearance of our cities. Such behavior confirms the doubts expressed before [12]. It seems that encouraging university workers to implement biometric tools for the purpose of architectural education will require a huge workload. It is important to explain that all social consultations, for example, those related to the perception of urban layouts, do not mean that the registered behaviors of non-professionals are supposed to be treated as final design decisions that should be implemented without criticism and consideration. The same issue in regard to the management of archaeological heritage was discussed by Brian J. Egloff [59].

*3.2. Eye Trackers in Architectural Education*

According to the studies conducted in other fields [45–54], it can be concluded that in general, architectural items are perceived in a different way by a non-professional, a student of the faculty of architecture, and a lecturer. However, conclusions from conducting such studies may be used in architectural education of all of the abovementioned groups. A graph was prepared in order to depict the potential applications of the eye tracker for education as well as to describe three perspectives mentioned in the title of the article (Figure 6).

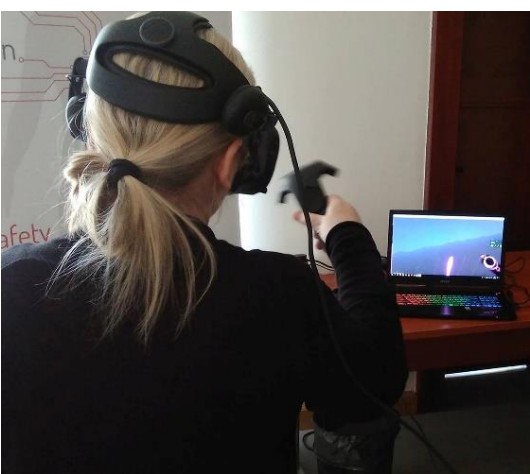

**Figure 5.** A university lecturer is shown the capabilities of an eye tracker and its software in the VR environment (photo from studies in 2018).

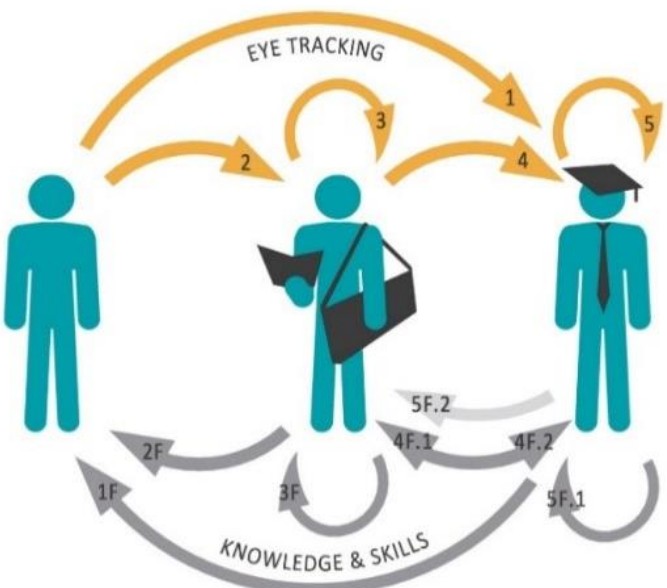

**Figure 6.** A graph presenting research-related relationships between the participants of eye tracking tests (by M. Rusnak and M. Rabiega).

### 3.3. Usefulness of Eye Tracking Research for Various Participants of the Education Process

Interdisciplinary character of architecture and its role are currently emphasized by experts, together with the importance of mutual communication not only between students and their teachers, but also investors and society [60]. Figure 6 presents the participants of the educational process in a simplified manner: a student, a lecturer, and a non-professional may all, to some extent, take part in the teaching process, here represented by eye-tracking tests. Such cooperation would most likely enrich the scientific environment with interaction/social participation. The graph may refer both to single individuals, e.g., ambitious students following personalized course of their studies, as well as teachers involved in their own original projects and groups: research clubs, those responsible for evaluating and modifying current curricula.

In order to systematize the graph, arrows were marked with colors and numbered. Each yellow arrow represents eye-tracking data obtained in specific situations. Grey arrows correspond to results, which can be obtained from the analysis of aggregated data. Detailed explanations are described below.

Test groups and examples of using eye trackers for scientific and teaching purposes are as follows:

1. Tests on non-professionals conducted by professionals constitute the most popular testing scheme.

2. Tests on non-professionals conducted by students—carried out, for example, within the scope of activity of research groups or workshops—are a chance for developing personal interests and introducing some individuals into the world of scientific research.

In favorable circumstances, as in the case of the discussed studies, 1 and 2 can be combined, and jointly published articles constitute the outcome of such cooperation.

3. Students' research during classes.

3.1. Group work—during classes and project tasks, students prepare the variants (as shown in Figure 3) and put forward hypotheses based on standard methods of analyzing, e.g., spatial arrangement [61], tacit knowledge [62] and/or intuitive opinion [63], and then verify them by testing fellow students. By presenting and discussing the results of their individual experiments referring to various aspects of the project, they are able to gain quite extensive experience in a short period of time.

3.2. Individual work—having access to an appropriately prepared eye-tracking laboratory, students develop and verify their knowledge based on programs prepared in advance [64]. In this case, tests may refer to, for example, the level of mastering the diagnosis of the reasons for corrosion in construction structures or the coordination of issues referring to various aspects of a project, analogically, to research done by the team managed by Mc Naughten, referring to the level of mastering resuscitation skills [47]. In this way, students can test their declarative and procedural knowledge on their own [65] or even monitor the level of their expertise [66].

4. Educators perform tests on future architects. Problems are diagnosed within the process of acquiring the knowledge by the students. It should, in particular, refer to laboratories, design studios, or even the needs of specially profiled postgraduate courses, such as conservation studies.

5. Diagnosis of lecturers' competencies. Educators wear an eye tracker while conducting classes or outside them, for example, during the performance of their professional duties. According to the definition by Chi, Glaser and Farr, one of the seven characteristics of an expert consists in the ability to reflect on their own activities, and an eye tracker would facilitate the self-monitoring process [67].

Potential benefits:

1F. Benefits for the society: Eye-tracking tests in the field of architecture and urban planning may influence the change of views on the perception of constructed structures, thus modifying the way of managing broadly understood urban space. Such research may initiate pro-social change in legislation referring to the construction activity as well as official procedures [68].

1F.1 and 2F.1. Benefits for the society: Making non-professionals involved in research would greatly support building social capital and trust [19]. As a result of opening to the voices of average citizens, it would be possible to stimulate care for common architectural space and a pro-social attitude to design in architects and future architects.

1F.2 and 2F.2. Benefits for participants: The results presented to non-professionals enable them to discover what they were focusing on during the tests, and thus learn more about the spontaneous activity of their sense of sight.

3F. Benefits for students.

3F.1. Group work: Students prepare the interpretations of the results of tests carried out on their classmates and observe whether they were able to foresee the results and whether the project meets their requirements and triggers the intended behaviors. According to the authors, such an approach may create the habit of devoting more attention to issues aimed at predicting the behaviors of users of designed space already at the stage of concept development, as well as promote understanding among different experts, e.g., designers of visual communication systems.

3F.2. Self-control and auto-corrections: Students are able to observe how well they have already mastered some activities and which aspects require more thorough work.

4F. Modification of the student–lecturer relationship.

4F.1. Student's perspective: The young learn what they are missing, to what they should devote more attention, and what form of cooperation brings the best results.

4F.2. Lecturer's perspective: Academics have a chance to learn which method of conducting their classes is the most effective.

5F. Lecturer's self-improvement.

5F.1. Personalized self-improvement: The experts, while interpreting the results of a test carried out on themselves, have a chance to begin the process of mastering their own teaching skills. What is important is that such tests do not have to refer exclusively to those beginning their scientific career. Thanks to them, senior lecturers could have the opportunity to feel more secure when approaching new generations. Eliminating obstacles in communication with students could result in more satisfaction.

5F.2. Student's perspective: Students would also benefit from the efforts undertaken by teachers, as the biggest obstacles in communication between these groups would be eliminated.

*3.4. Influence on the System of Management of Architectural Education*

By combining, in a synergetic way, the results of eye-tracking tests performed on different groups, it will be possible, with full knowledge of the scale and scope of issues that emerge, to undertake a re-modeling of the curricula in force, both with reference to specific subjects, as well as to the entire course of studies. Thanks to the combination of eye-tracking research with carefully thought-out interviewing, persons responsible for preparing university curricula would receive feedback on whether their perception of the assets of a given offer is seen in the same way by the addressees, i.e., students. Before undertaking the activities described above, it is crucial to clearly define the aim and scope of tests as well as to adopt the appropriate methodology for conducting the experiments in order to be able to juxtapose and compare them. Only thanks to a carefully designed process of data collection will it be possible to obtain reliable results enabling the monitoring of changes.

Importantly, the introduction of classes with the use of eye trackers into university curricula is possible and desired, even in the view of the current wording of the ordinance on the curricula of architectural studies. The Ordinance of the Minister of Science and Higher Education from the 18th of July 2019 on the standards of education preparing for performing the profession of an architect [11] allows a university to devote at least 405 hours for conducting classes of the university's own choice for undergraduate studies, and at least 145 hours for postgraduate studies. The university is free to offer any classes as long as they complement the knowledge, skills or social competencies of the students. Eye-tracking research undoubtedly enables the broadening of many qualifications specified in the act and the authors believe that such research may be beneficial since the curriculum clearly does not include enough field or practical studies (apart from the obligatory internship), which would make it possible to confront design assumptions with reality, not to mention that the implementation of innovative solutions applied by the researchers in their studies would surely create great interest as is the case at the Faculty of Architecture of Wrocław University of Science and Technology. Appropriately prepared classes conducted with the use of the eye-tracking technique may perfectly translate into various learning results, divided into general and detailed, and they, in turn, would most likely bring positive effects in the context of knowledge, skills and social competencies. Classes with the use of an eye tracker may lead to a better understanding of the relationship between humans and architecture as well as between architecture and the surrounding environment; they could also emphasize the necessity to adjust architecture to human needs and the human scale (general result; scope–knowledge).

What is the most important, students—as future architects—managed to learn how to diagnose problems and "translate them into valid and accurate questions" [69] not by designing, but by conducting experiments. They proved able to answers some of those questions on their own, while others will require future discussions with experts from other fields [69].

## 4. Discussion

### 4.1. Potential of Implementing Eye Tracking in University Architectural Education

The introduction of eye-tracking studies into the curricula of architecture and related studies is connected with a number of advantages, which include the following:

- An innovative way of directing the attention of future architects to the issue of order in architecture and urban planning through broadening their knowledge of perception of architecture, i.e., how to design in order to attract one's gaze, while at the same time, correctly inscribing one's design into the natural or historical context;
- Boosting the students' interest in the experimental aspect of architectural research, which may translate into solving design challenges in a more creative way;
- Broadening the students' social and technological competencies, which makes them able to accept non-standard and complex design orders in the future that will require in-depth analysis of visual needs of the users as well as interdisciplinary cooperation;
- Self-monitoring of lecturers and students;
- Positively influencing the student–lecturer relationship, which should result in the increased number of those willing to continue their personalized career-development path during doctoral studies;
- Promotion of the university: Distinguishing the offer of the university from other research centers, both by using technologically advanced solutions as well as by matching teaching requirements to actual needs;
- Bringing the topic of the perception of architecture close to its everyday users, making them interested in the buildings that they pass by every day and promoting the profession of the architect.

However, the suggested solution also includes the following disadvantages:

- The high cost of purchasing eye-tracking equipment together with the necessity to include in the budget of a research unit the expenses for maintenance and conservation of such devices as well as the insurance of eye trackers due to their high value;
- The necessity to prepare an entire laboratory—one device can be used by a limited number of students, and project groups usually consist of a dozen members;
- Lecturers may question the legitimacy of using the method for their self-analysis, as it requires them to devote additional time and effort as well as to adopt an open-minded and self-critical attitude;
- Classes need to take place at university, so in such exceptional conditions as the COVID-19 pandemic, the intended use of eye trackers during classes is impossible.

The authors are aware that most of what is listed above requires confirmation. However, the aforementioned research on the education of specialists in other fields—pedagogy, medicine (pediatrics, resuscitation, laboratory diagnostics), criminology, car transportation, aviation or shipping [45–52] in combination with numerous cited research endeavors regarding landscape, urban planning, architecture, museum studies and art [26–42]—seem to justify making promising assumptions about the usefulness of eye tracking in architectural education.

### 4.2. Eye Tracking and Other Methods of Measuring People's Reactions

An eye tracker determines the spots we look at but tells us little about the motivation of the observers. This refers in particular to registrations of behaviors in situations where the observers are given complete freedom and do not need to carry out any specific task [70].

The authors are aware that there are other, commonly used ways of researching the opinion of the public, such as surveys or interviews, but their application in a didactic

process is fairly unattractive. A very attractive method of diagnosing emotions is to observe the participants engaged in experiments making use of virtual reality projections. On the other hand, science can offer many more advanced tools, making it possible to measure people's physiological reactions. However, at the current level of development, they are much more invasive than eye trackers. Probably few lecturers or students would agree to participate in a class while wearing an EEG cap. Eye trackers seem to be a perfect solution. What is important is that the use of eye trackers does not exclude employing other methods. Most often, eye trackers will yield the best results when combined with a survey. It is also possible to join an eye-tracking registration with measuring pulse, skin conductance or a video recording of the participant's facial expressions during the test [71]. When introducing new tools into an educational process, it is important to make sure that both the students and the lecturers will be able to understand how such a device works, how to operate it and interpret the results. Combining several techniques and tools for scientific purposes seems very attractive, but eye tracking, which allows direct registering of visual reactions on presented stimuli (photos, videos) appears to be the most available and suitable option for future architects. Due to being disconnected from visual stimuli, data related to brain activity or graphs showing changes in skin conductance seem to be a less attractive addition to the university curriculum. The more complex the methodology that is used, the less likely it becomes to achieve a realistic diagnosis of the educational effect.

## 5. Conclusions

After four years of cooperation and research, the authors have no doubts that eye trackers should in the future find broad application in the field of educating architects and urban planners, as they make it possible to record what attracts our attention or distracts the users of specific spaces, how people perceive different elements defining squares, streets and passages, and therefore what improves their orientation in the city as well as inside buildings. This tool, if used in an appropriate way, would be useful for lecturers, students, and non-professionals. It would make it possible to broaden their knowledge in the field of perception of architecture by ordinary people, verify project assumptions, make students gain knowledge on experimental research as well as promote their self-improvement and self-development of teachers.

Architecture, just like the people who design it, undergoes numerous transformations. The methods of architectural education should be adjusted to match such changes as quickly as possible. It might prove very difficult for architects to successfully implement such changes and transformations in their designs if they know little about how regular people perceive build environments. How can one respond to the expectations of people with special needs, e.g., preschool children or Alzheimer's patients, without knowing how they see their surroundings? That is why the education of future designers should include elements of experiments that might show them how important it is to recognize the complex relationship between neuroscience and architecture [72].

**Author Contributions:** Conceptualization, M.A.R. and M.R.; methodology, M.A.R. and M.R.; software, M.A.R. and M.R.; validation, M.A.R. and M.R.; formal analysis, M.A.R. and M.R.; investigation, M.A.R. and M.R.; resources, M.A.R.; data curation, M.A.R. and M.R.; writing—original draft preparation, M.A.R. and M.R.; writing—review and editing, M.A.R. and M.R.; visualization, M.A.R. and M.R.; supervision, M.A.R.; project administration, M.A.R.; funding acquisition, M.A.R. All authors have read and agreed to the published version of the manuscript.

**Funding:** This research received no external funding. Financial support comes from the place of authors employment, Wrocław University of Science and Technology.

**Institutional Review Board Statement:** Not applicable.

**Informed Consent Statement:** Informed consent was obtained from all subjects involved in the study.

**Data Availability Statement:** The analyzed data is contained in the quoted articles and their data extensions.

**Conflicts of Interest:** The authors declare no conflict of interest.

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
