# Peer review of "The Potential of Using an Eye Tracker in Architectural Education: Three Perspectives for Ordinary Users, Students and Lecturers"

_buildings, doi:10.3390/buildings11060245_

Round 1

Reviewer 1 Report

This manuscript presents a proposal methodology that is based on the use of ewe tracker tools in support of architectural design developments.

The proposal and concept are of interest for the Buildings' readers, and should be considered for publication. The use of technology in design procedures seems promising and will manifest in future developments of high impact.

Regarding the current manuscript, however, the presentation is still weak and lacks of important content and outcomes.

For example:

  • at least a short review on available tools compared to the proposed solution should be recalled
  • the proposal lacks of a practical application / experimental investigation that could be used to support the concept
  • the methods and potentials of the approach are not sufficiently described and commented in the paper
  • the state of art does not reflects the actual advancements on the topic
  • more qualitative and quantitative analysis of data is necessary. Otherwise, the document itself does not support properly the methodology that is only roughly introduced in the manuscript

Author Response

Thank you for all your comments.
It helped us to refine the article. We tried to comply with all comments.

The text has been checked by a professional translator. We expanded and updated the eyetracking research we refer to.
Any fragments added have been marked in blue in the new version of the text.

Yours faithfully
Marta Rusank and Mateusz Rabiega

Reviewer 2 Report

The Abstract is too convoluted, with overly long sentences. Please re-write with concise statements and include results, not elaborate on statement of process. Relevant references are missing on the contradiction between popular and professional preferences in design. Pleas add:

https://forum.savingplaces.org/blogs/special-contributor/2019/09/26/building-tomorrows-heritage-architectural-myopia

https://www.shareable.net/architectural-myopia-designing-for-industry-not-people/

More basic references on eye-tracking:

Ann Sussman & Justin B. Hollander (2021) Cognitive Architecture, 2nd Edition, Routledge, New York, USA.

Nikos Salingaros & Ann Sussman (2020) “Biometric pilot-studies reveal the arrangement and shape of windows on a traditional façade to be implicitly ‘engaging’, whereas contemporary façades are not”, Urban Science, Volume 4, Issue 2: article 26, 1-19.

I'm not sure that Figs 2 & 3 are necessary in a scientific paper.

Although the topic addressed in this paper is extremely important for the future of architectural design, the present paper is incomplete. The authors spend all their time discussing the project, the details of which are of only secondary interest. The striking results that other authors have obtained using eye tracking are not presented here. Even the building shown in figure 5 is not evaluated, something that would carry a strong message for the reader.

If the authors do not have actual data to demonstrate the utility of eye tracking, then they should refer to the results of other authors. As the paper remains in its present form, it simply promises a lot, but delivers very little. However, the authors evidently know the method, and understand its potential. I am sure they will be able to fill-in the material required to make this a really good paper.

Author Response

Thank you for taking the time to review our article. The comments we received allowed us to greatly improve the text. We are grateful for pointing out specific articles. We have modified the type and number of attached illustrations. Our manuscript was finally checked by a qualified translator.

 To facilitate a re-review, all newly added fragments have been marked in blue.

Greetings
Marta Rusnak and Mateusz Rabiega

Round 2

Reviewer 1 Report

The original paper has been largely revised and improved, compared to the original version, and this is appreciated.

However, it is still not clear the research method.

  • figure 2: which kind of software has been used to obtain it? It is not described in the manuscript
  • what's the difference between figures 3 and 5?
  • students are cited in the text, it seems they have been reasonably involved as volunteers for the trials and experiments. Did the authors involved a group of volunteers? How many? Gender? Age? All these data in support of the experimental analysis are missing. As such, experimental methods cannot be considered properly described. A scientific approach seems missing. I am confident it is not so, but the manuscript in this current version does not give  evidence of a clear scientific approach but presents some examples of techniques, etc. please work on this side to give scientific value to this document and behind study
  • following the previous comments, it is not clear the final goal of this document. Is it to compare the potential of research methods that are based on eye tracking? Thus a summary of available techniques? And thus present something close to a "review" of available techniques? Or other? There's a strong difference in these approaches, and also a totally different structure / content of the document that should be revised accordingly

Author Response

The original paper has been largely revised and improved, compared to the original version, and this is appreciated.

However, it is still not clear the research method.

-The paper is written on the basis of a review of available literature and the authors’ own experiences. Available methods and techniques have been analyzed from the perspective of using eye trackers in architectural education. Following the suggestions of reviewers, the text has been enriched with references to experts who comment on the detachment of architects from the needs of ordinary users, which turns the paper into a sort of a manifesto.

figure 2: which kind of software has been used to obtain it? It is not described in the manuscript

-Thank you for this comment. The description has been improved.

what's the difference between figures 3 and 5?

-Thank you again. The previous version might have been misleading. In the current version we have combined both figures – the first showed resources used in research that was published afterwards, while the second displayed other ideas prepared by the students, which were not be used in eye-tracking research due to time limitations and lack of equipment but are worth mentioning because they show doubts and questions that the students came up with, e.g. to what extent can architects and conservators diagnose the perception of monuments.

students are cited in the text, it seems they have been reasonably involved as volunteers for the trials and experiments. Did the authors involved a group of volunteers? How many? Gender? Age? All these data in support of the experimental analysis are missing.

-The paper states that a total of 51 student had a chance to use an eye tracker (the color-marked fragment of part 3. Results) Information about their age has been added. Moreover, the characteristics of the group – number, age, sex – are given in each referenced paper.

As such, experimental methods cannot be considered properly described. A scientific approach seems missing. I am confident it is not so, but the manuscript in this current version does not give  evidence of a clear scientific approach but presents some examples of techniques, etc. please work on this side to give scientific value to this document and behind study.

Following the previous comments, it is not clear the final goal of this document. Is it to compare the potential of research methods that are based on eye tracking? Thus a summary of available techniques? And thus present something close to a "review" of available techniques? Or other? There's a strong difference in these approaches, and also a totally different structure / content of the document that should be revised accordingly

-The research is based on the presentation of an observed issue – its goal was to determine its importance on the basis of literature studies, in which similar discrepancies have been noted. Such study shows that what we might call architectural shortsightedness is a common phenomenon and requires a modified system of architectural education. Eye tracking appears to be one of possible solutions. Since eye tracking has not been introduced in architectural education yet, our deliberations progressed in two ways. On the one hand, we possessed certain experience that we expanded by showing other examples of how eye trackers are used for architectural diagnosis. On the other hand, we reached for examples of how eye trackers are used in education, which show possible effects of such diagnosis. A combination of conclusions from those two approaches allowed us to suggest that eye trackers have the potential to be a solution to the abovementioned issue. 

Thank you for your time. Each remark enriches our article.

Reviewer 2 Report

The revised version is much improved and should be published. 

Author Response

Thank you for your time. Your remarks enriches our article. 

Round 3

Reviewer 1 Report

The paper has been further revised, and there are some improvements in the presentation of methods. On the other side, the paper still lacks of appropriate input. Minor details are required, but they can make the difference.

Eye tracker tools can support building design. There are already several studies. A dedicated section of the document could at least cite the use of more complex virtual tools that are not based only on eye tracking but also on further artificial intelligence tools. This is line with the study reported in:

Buildings 2021, 11(5), 204; https://doi.org/10.3390/buildings11050204 The used software allows also for eye tracking, but additional action units. Do the Authors believe this approach can be useful for their proposal and methodology? Given that there are several commercial software packages on the market, it would be important (at least) to comment the potentials and future applications in the architectural and building context. The caption of figure 2 is now appropriate in content, but not in form. Is is surely too long for a figure caption. Please move part of the description to the main text and use appropriate words to describe it. Is the same figure taken from a literature document? It is not clear if this figure results from an original research trial of the current study and manuscript, or not.

Author Response

The paper has been further revised, and there are some improvements in the presentation of methods. Minor details are required, but they can make the difference.

Eye tracker tools can support building design. There are already several studies. A dedicated section of the document could at least cite the use of more complex virtual tools that are not based only on eye tracking but also on further artificial intelligence tools. This is line with the study reported in:Buildings 202111(5), 204; https://doi.org/10.3390/buildings11050204 The used software allows also for eye tracking, but additional action units. Do the Authors believe this approach can be useful for their proposal and methodology?

-Thank you. Our intention was to focus on eye tracking. We mention other techniques in the paper and now justify why we deem this particular group of devices the most suitable for the purposes of enriching architectural education. We have increased the number of references indicating to the reader that eye tracking is one of possible methods allowing a widening of the interdisciplinary competences of future architects. Although the previous version of the paper already mentioned the possibility to diagnose emotions by means of the method applied in the paper suggested by the reviewer, we did not include a specific reference in the bibliography then. Thank you once again for mentioning an interesting paper that we had not read before.

Given that there are several commercial software packages on the market, it would be important (at least) to comment the potentials and future applications in the architectural and building context.

-The paper mentions some specific names and we included references to papers that deal with that issue during the previous stage of revision.

The caption of figure 2 is now appropriate in content, but not in form. Is is surely too long for a figure caption. Please move part of the description to the main text and use appropriate words to describe it. Is the same figure taken from a literature document?

-Thank you, we have changed this aspect.

It is not clear if this figure results from an original research trial of the current study and manuscript, or not.

-It has been explained. Thank you very much for your time and for pointing out the different drawbacks of our paper.